# Spongin as a Unique 3D Template for the Development of Functional Iron-Based Composites Using Biomimetic Approach In Vitro

**DOI:** 10.3390/md21090460

**Published:** 2023-08-22

**Authors:** Anita Kubiak, Martyna Pajewska-Szmyt, Martyna Kotula, Bartosz Leśniewski, Alona Voronkina, Parvaneh Rahimi, Sedigheh Falahi, Korbinian Heimler, Anika Rogoll, Carla Vogt, Alexander Ereskovsky, Paul Simon, Enrico Langer, Armin Springer, Maik Förste, Alexandros Charitos, Yvonne Joseph, Teofil Jesionowski, Hermann Ehrlich

**Affiliations:** 1Faculty of Chemistry, Adam Mickiewicz University, Uniwersytetu Poznańskiego 8, 61-614 Poznan, Poland; markot6@amu.edu.pl (M.K.); barles5@amu.edu.pl (B.L.); 2Center of Advanced Technology, Adam Mickiewicz University, Uniwersytetu Poznańskiego 10, 61-614 Poznan, Poland; mpszmyt@amu.edu.pl; 3Institute of Electronic and Sensor Materials, TU Bergakademie Freiberg, Gustav-Zeuner-Str. 3, 09599 Freiberg, Germany; voronkina@vnmu.edu.ua (A.V.); parvaneh.rahimi@esm.tu-freiberg.de (P.R.); sedigheh.falahi@doctorand.tu-freiberg.de (S.F.); yvonne.joseph@esm.tu-freiberg.de (Y.J.); 4Department of Pharmacy, National Pirogov Memorial Medical University, Vinnytsya, Pyrogov Street. 56, 21018 Vinnytsia, Ukraine; 5Institute of Analytical Chemistry, TU Bergakademie Freiberg, Leipziger Str. 29, 09599 Freiberg, Germany; korbinian.heimler@chemie.tu-freiberg.de (K.H.); anika.rogoll@chemie.tu-freiberg.de (A.R.); carla.vogt@chemie.tu-freiberg.de (C.V.); 6IMBE, CNRS, IRD, Aix Marseille University, Station Marine d’Endoume, Rue de la Batterie des Lions, 13007 Marseille, France; alexander.ereskovsky@imbe.fr; 7Max Planck Institute for Chemical Physics of Solids, Nöthnitzer Str. 40, 01187 Dresden, Germany; simon@cpfs.mpg.de; 8Institute of Semiconductors and Microsystems, TU Dresden, Nöthnitzer Str. 64, 01187 Dresden, Germany; enrico.langer@tu-dresden.de; 9Department Life, Light & Matter, University of Rostock, Albert-Einstein-Str. 25, 18059 Rostock, Germany; armin.springer@med.uni-rostock.de; 10Medical Biology and Electron Microscopy Centre, Rostock University Medical Center, Strempelstr. 14, 18057 Rostock, Germany; 11Institute for Nonferrous Metallurgy and Purest Materials (INEMET), TU Bergakademie Freiberg, Leipziger Str. 34, D-09599 Freiberg, Germany; maik.foerste@inemet.tu-freiberg.de (M.F.); alexandros.charitos@inemet.tu-freiberg.de (A.C.); 12Faculty of Chemical Technology, Institute of Chemical Technology and Engineering, Poznan University of Technology, Berdychowo 4, 60-965 Poznan, Poland; teofil.jesionowski@put.poznan.pl

**Keywords:** *Hippospongia communis*, sponge, biocorrosion, lepidocrocite, sensor, dopamine, biomineralization, biomaterials, biomimetics

## Abstract

Marine sponges of the subclass Keratosa originated on our planet about 900 million years ago and represent evolutionarily ancient and hierarchically structured biological materials. One of them, proteinaceous spongin, is responsible for the formation of 3D structured fibrous skeletons and remains enigmatic with complex chemistry. The objective of this study was to investigate the interaction of spongin with iron ions in a marine environment due to biocorrosion, leading to the occurrence of lepidocrocite. For this purpose, a biomimetic approach for the development of a new lepidocrocite-containing 3D spongin scaffold under laboratory conditions at 24 °C using artificial seawater and iron is described for the first time. This method helps to obtain a new composite as “Iron-Spongin”, which was characterized by infrared spectroscopy and thermogravimetry. Furthermore, sophisticated techniques such as X-ray fluorescence, microscope technique, and X-Ray diffraction were used to determine the structure. This research proposed a corresponding mechanism of lepidocrocite formation, which may be connected with the spongin amino acids functional groups. Moreover, the potential application of the biocomposite as an electrochemical dopamine sensor is proposed. The conducted research not only shows the mechanism or sensor properties of “Iron-spongin” but also opens the door to other applications of these multifunctional materials.

## 1. Introduction

Marine sponges are a resourceful provider of a large diversity of biologically active compounds and biological materials, including chitin and spongin [1,2]. Proteinaceous spongin in the form of 3D porous network-like scaffolds is recognized as a renewable marine biomaterial due to its ability of selected demosponges (mostly bath sponges) to grow under marine ranching conditions [3]. It consists mainly of protein-derived collagen of still unknown type [4], a significant amount of sulfur (up to 5%) similar to keratins, unique halogenated amino acids, xylose, as well as traces of calcium carbonates and silica [5,6,7,8]. This biopolymer is characterized by a complex hierarchical structure based on interconnected nano- and micro-fibers [9,10,11,12]. Such a composition gives this marine biomaterial special resistance to a wide range of acids and enzymes as well as specific structural and mechanical features [13]. Consequently, there are numerous fields of spongin’s applications in the form of ready-to-use scaffolds, including tissue engineering and biomedicine [14], as well as bioinspired material science [15,16,17,18].

In addition, spongin’s range of applications in extreme biomimetics [19] is enhanced due to its thermal stability of up to 300 °C [6]. Three-dimensional spongin scaffolds can also be carbonized at high temperatures under anaerobic conditions. Carbonization at 1200 °C confirms the preservation of spongin scaffold morphology in the formed graphite [20]. All these features mark a breakthrough opportunity in modern materials science with respect to spongin-based scaffolding strategies [20,21,22,23,24,25,26,27,28,29,30,31].

Biomimetics is the science-driven imitation of the natural phenomena, processes, and fascinating architectural principles of natural materials using a wide range of modern tools [22]. It is an interdisciplinary direction in the creation of new materials with unique properties for broad practical applications, where special priority is given to renewable biopolymers such as spongin, which precludes the deliberate depletion of natural resources. By combining inorganic compounds (e.g., iron (III) chloride [20], titanium (IV) oxide [24], manganese (IV) oxide [23], and copper (II) tetraamine chloride [19,22]) and spongin using a nature-inspired biomimetic approach, it can provide highly attractive solutions to current technological challenges and lead to the development of new advanced, sustainable, and biodegradable composite materials [32].

Intriguingly, the skeletons of selected species of spongin-based bath sponges represent examples of naturally occurring iron-containing 3D composites formed due to the corrosion of artificial iron constructs found in marine environments (Figure 1). The biocorrosion of metal structures in seawater is the cause of elevated iron ion concentrations in water [33]. Consequently, iron ions are involved in biomineralization, which results, as an example, in the formation of a unique iron-based mineral phase, lepidocrocite, on the organic part of the three-dimensional skeleton of the marine sponge–sponging (see for an overview [5]) (Figure 1). Crystalline lepidocrocite (γ-FeOOH) is an iron oxide–hydroxide mineral with magnetic properties [34]. It is red to reddish-brown in color and has a sub-metallic luster. Lepidocrocite is commonly found on rusted metal structures underwater, in soils, and in iron ore deposits [35,36]. It is stable over a wide range of temperatures (10–60 °C) and pH (4.0–8.0) [37]. Previously, lepidocrocite as a mineral was applied as a sensor, catalyst [38,39,40], and adsorbent of diverse pollutants [37,41] and pigments [42].

Diverse methods for the synthesis of lepidocrocite without the presence of organic matrices were proposed previously. For example, this mineral phase can be obtained by the oxidation of FeCl_2_ with NaClO_3_ under slow heating of a common solution from 20 to 75 °C [43]. The “Process for the preparation of synthetic lepidocrocite” where this compound “can be produced by reacting an aqueous iron (II) chloride solution with aqueous alkali metal hydroxides with simultaneous oxidation with atmospheric oxygen” has been patented [44].

In this study, inspired by the previously reported phenomenon of natural biomineralization of iron into lepidocrocite in demosponges [45,46,47], a biomimetic method for the development of lepidocrocite on spongin scaffolds using artificial seawater under laboratory conditions, is proposed. The reaction in an artificial seawater environment between a spongin template and iron ions is presented, which leads to the formation of a new 3D composite material called “Iron-Spongin” that resembles the size and shape of the original sponge skeleton. The corresponding mechanism for the possible formation of crystalline lepidocrocite on spongin is discussed. This simple biomimetic approach has led to obtaining a specific multifunctional material that can be readily fabricated with realistic prospects for large-scale application within the framework of the marine bioeconomy of sponges [2]. Moreover, for the first time, a potential application of this unique lepidocrocite-spongin composite as a sensor for dopamine (DA) detection is proposed.

## 2. Results

### 2.1. Confocal Micro X-ray Fluorescence (CMXRF)

Preliminary experiments with the aim to obtain knowledge of the chemistry of naturally occurring rusty sponges were carried out using CMXRF techniques. Thus, corresponding measurements were performed for the samples of spongin scaffold with naturally formed lepidocrocite (“Spongin Fe-natural”) and the control sample of the spongin scaffold (“Spongin pure”) (Table 1) with identical measurement parameters.

From the fluorescence spectra of sample ‘Spongin Fe-natural’ five main elements are identified: sulfur, calcium, iron, bromine, and iodine. All five elements are assigned to the spongin structure (Figure 2A). Due to the relatively high count rates for iron (Table 1), a representative 3D distribution image for this element could be generated, which is in very good structural agreement with the video image (view on the top) provided by the spectrometer camera (Figure 2A).

For all the other four observed elements, far more diffuse elemental distribution images are obtained, caused by the overall lower signal count rates (Table 1). Nevertheless, the quality of the 3D reconstructions still allows assigning these elements to the spongin structure (Figure 2A). Even the distribution of S shows some correlation with the distribution of the other elements, especially when measured at high sample densities (e.g., at conjunctions of the sponge strings). This result was rather unexpected since S with a relatively low Z number exhibits the lowest lateral resolution of the 5 elements detected in this sample (diameter of probing volume approx. 69.0 µm) together with a low fluorescence yield due to the high liability for absorption effects.

The same five elements and, additionally, silicon are identified from the fluorescence spectra of the control sample ‘Spongin pure’. Only four of them (sulfur, calcium, iodine, and bromine) can be assigned to the spongin structure. Hereby, in contrast to the ‘Spongin Fe-natural’ sample, the most representative reconstruction of the spongin structure provides a 3D distribution image of bromine (Figure 2B, green). This is due to the relatively high fluorescence energy of bromine (Br Kα: 11.902 keV) and the coherent smaller excitation volume. The iron distribution (Figure 2B, red) for the control sample can also be assigned to the sponge structure, but it does not show a homogenous distribution throughout the sample and is distributed rather pointwise, and the absolute signal count rate for Fe in the control sample (compared to the ‘Spongin Fe-natural’ sample) is also much lower (Table 1). However, for all the observed elements, diffuse elemental distribution images (Figure 2B) were obtained. In particular, intensified silicon fluorescence radiation can be detected from a certain spot in the sample (Figure 2B, cyan). By matching it with the bromine distribution pattern, it seems to be located within the spongin structure and is probably a grain of sand (quartz) incorporated into the spongin structure (see example [48]).

### 2.2. Digital Microscopy

In the images obtained with a digital microscope (Figure 3), a significant difference was observed in the appearance of the control sample and the “Iron-Spongin” sample after the ultrasound treatment. After 30 days of initiated corrosion, the spongin acquired a consistent rusty color, indicating the transformation of iron powder into an iron oxide form that was tightly bound to the organic matter.

### 2.3. Scanning Electron Microscopy (SEM) with Energy Dispersive X-ray Analysis (EDX)

The SEM images in Figure 4 show the control sample and “Iron-Spongin” after ultrasonic treatment. In Figure 4A,B, a network of spongin microfibers is observed, which forms complex porous formations. An analysis of the SEM images confirmed that uniform deposition of iron oxide crystals occurred during the initiated corrosion. The SEM images in Figure 4C,D show spongin fibers densely covered with iron oxide clusters. In the approximation in Figure 4E,F, crystal-like structures can be clearly observed. The high quality of the inorganic coating may be due to the corrosion-initiated synthesis procedure, which took 30 days. Importantly, the unique porous structure with numerous iron oxide clusters was preserved even after ultrasonic treatment for 2 h.

EDX analysis performed on an “Iron-Spongin after ultrasound treatment” material in the area with visible crystal-like structures showed a very high iron content (34.4 at%). In a control sample of spongin in seawater, the iron content was detected to be very low (0.2 at%). This confirmed the formation of crystals during the initiated corrosion, consisting mainly of iron (Figure 5) (for details, see also Appendix A). The distribution of elements within the spongin fibers is shown in Figure 6 and Figure 7. The results of the biomineralization are well visible both in the longitudinal (Figure 6A) and in the cross-section (Figure 6B and Figure 7) of the fiber as two different (inner and outer) layers. The differences in the content of Fe and O in these layers are also noticeable (Figure 7).

### 2.4. High-Resolution Transmission Electron Microscopy (HR-TEM)

HR-TEM analysis was used to confirm the presence of crystalline phase as lepidocrocite in the “Iron-Spongin after ultrasound treatment” sample. Figure 8A shows a cross-section of a selected section of the composite fiber with lath-like Fe-containing nanoparticles forming conglomerates inside the outer shell of the spongin fiber. This indicates the effective binding of the iron-containing phase to spongin during biomimetic-initiated corrosion under laboratory conditions. The HR-TEM image shows that the particles consist of several nanocrystallites with crystallite sizes of 3–5 nm (Figure 8B). The calculated FFT of the HR-TEM image of the “Iron-Spongin after ultrasound treatment” sample consists of discrete diffraction spots of randomly oriented nanocrystals, reflecting the fine crystallinity of the particles (Figure 8C). An analysis of the reflections indicates that the majority of the particles can be attributed to the orthorhombic lepidocrocite phase (Amam space group [36] or Cmcm [49]). There are also interplane separations of 1.55 Å, 2.64 Å, and 2.66 Å consistent with the (110) and (100) planes of hexagonal feroxyhyte [50]. It is an unstable aqueous iron oxide that transforms spontaneously into goethite and is usually formed under high-pressure conditions on the ocean grounds [51]. For example, according to Vacelet and co-workers, lepidocrocite and a small amount of goethite are minerals that occur in the natural iron-rich skeletons of spongin-based *Spongia officinalis* marine demosponges [46].

### 2.5. Fourier-Transform Infrared Spectroscopy

In an attempt to identify the ferrous layer formed on the spongin scaffold under study, FTIR spectroscopy of the materials was performed to examine the presence of characteristic functional groups. Detailed studies were carried out for spongin control samples in seawater as well as “Iron-spongin”, before and after ultrasound treatment (Figure 9A). Additional measurements were made for iron powder after 30 days in the seawater with and without the presence of the spongin scaffold (Figure 9B) (details of the bands present in the spectra, with their wave numbers and band assignments, are given in Appendix A).

Most of the bands in the FTIR spectra of “Iron-Spongin” and “Iron-Spongin after ultrasound treatment” correspond to the bands in the spectrum of the control sample of spongin in seawater. The bands that occur only in the samples in the presence of corroded iron powder are 570, 740, 1021, and 1150 cm^−1^ (Figure 9A). The band at 570 cm^−1^ is characteristic of Fe-O vibrations in iron oxides [52,53]. The most intense band at 1021 cm^−1^ in the FT-IR spectra is associated with lepidocrocite (γ-FeOOH) [54]. The bands at 1150 cm^−1^ and 740 cm^−1^ can also be assigned to OH deformation and bending in γ–FeOOH [55]. The high-intensity bands at 1021 cm^−1^ and 740 cm^−1^ may suggest that a well-crystallized lepidocrocite phase is strongly present.

The effect of spongin scaffold on iron corrosion in seawater was also investigated (Figure 9B). The band characteristic of iron oxides (570 cm^−1^) and lepidocrocite (740, 1021, 1150 cm^−1^) were observed only in the FTIR spectrum of the corroded iron powder after 30 days in seawater in the presence of the spongin scaffold.

### 2.6. X-ray Diffraction

The X-ray diffraction pattern of spongin under study is similar to that reported previously [15,22,24]. The treatment of spongin samples with iron powder using artificial seawater (see Section 4) causes the appearance of reflection characteristics for that of lepidocrocite [56], which confirms that this mineral phase is formed during the preparation of the “Iron-Spongin” composite. Further data analysis confirms the formation of lepidocrocite on both the “Iron-Spongin” sample and on “Iron-Spongin after ultrasound treatment”. This is indicated by the peaks present in the XRD graphs of these samples at ~14°, ~27°, ~38°, ~47°, ~53°, ~61°, and ~68°, which correspond to the (020), (120), (111), (020), (151), (231), and (251) crystal planes, respectively (Figure 10). These peaks correspond to polymorphs of the iron oxyhydroxide lepidocrocite (γ-FeOOH) [57]. For comparison, a diffractogram of iron powder (Figure 10E) obtained after 30 days in seawater in the presence of spongin with lepidocrocite-characteristic reflections is included.

### 2.7. Thermogravimetric Analysis

The thermal degradation of a control spongin sample in seawater and “Iron-Spongin after ultrasound treatment” was studied. Two weight losses occur during the thermal degradation of both samples (Figure 11). The first, a weight loss of about 5–8% in the 70–150 °C range, is related to the evaporation of physically adsorbed and hydrogen-bonded water from the spongin scaffold [12]. The second weight loss in the temperature range of 230–450 °C is about 63.2% for the control sample and about 44.8% for “Iron-Spongin after ultrasound treatment”. This may be related to the decomposition of the protein matrix: the disintegration of the peptide bonds [12,25], and thermal degradation of disulfide bonds [12,58] and hydrogen bonds [12]. In the “Iron-Spongin after ultrasound treatment” material, the thermal stability is higher than that of the control spongin sample. The difference in thermal stability can be attributed to the formation of bonds between spongin and iron and electrostatic interactions formed between the hydroxyl groups of spongin and lepidocrocite [29].

### 2.8. Magnetic Properties

The “Iron-Spongin after ultrasound treatment” is attracted by a neodymium magnet with a pull force of 192 N (see Appendix A). It is well known that lepidocrocite is paramagnetic at room temperature with low field magnetic susceptibility [34,59,60]. Paramagnetism is the phenomenon whereby a material magnetizes in an external magnetic field in a direction consistent with the direction of the external field. This phenomenon occurs in all atoms and molecules with unpaired electrons, e.g., free atoms, free radicals, and transition metal compounds that contain ions with unfilled electron shells [61]. Paramagnetic materials have a relative magnetic permeability slightly greater than 1 (i.e., low positive magnetic susceptibility), and are therefore attracted to magnetic fields [62]. In contrast to the “Iron-Spongin after ultrasound treatment” represented here, all 35 naturally occurring rusty sponges (see Figure 1B,D and Figure 15), which are approved for their magnetic features under the same experimental conditions, show no attachment to the neodymium magnet.

### 2.9. Dopamine Detection

The application of spongin-based sensors remains to be in trend. In this study, we used the developed composite for the detection of DA. This compound is a vital catecholamine neurotransmitter found in mammals’ central and peripheral nervous systems. It regulates a wide variety of neuronal functions, including emotion, behavior, cognition, learning, memory, and movement. In living systems, DA controls the transmission of signal messages to various domains of the brain and other parts of the body. In addition, DA receptors are vital targets for neuropsychiatric illnesses such as depression, Parkinson’s, schizophrenia, and Huntington’s [63,64]. Therefore, the quantitative detection of DA in biological and chemical systems is critical. Various analytical methods are used for the detection of DA, but each of them has some disadvantages. Among them, electrochemical methods have proven to be the most effective for the determination of DA in the presence of other biological molecules [65,66,67,68]. However, developing a simple, cost-effective, and compatible composite material as an electrode material for the selective detection of DA at low concentrations without interfering with other biologicals is challenging.

Herein, for the first time, a novel, low-cost, sensitive, and selective electrochemical sensor for the detection of DA based on carbon paste electrodes (CPE) modified with naturally occurring iron-spongin and biomimetic “Iron-Spongin after ultrasound treatment” is developed. The electrodes are denoted as N-Iron-Sp/CPE and B-Iron-Sp/CPE, respectively. The amperometric responses of N-Iron-Sp/CPE and B-Iron-Sp/CPE for the successive addition of different concentrations of DA in 0.1 M phosphate buffer pH 6.5 are given in Figure 12A. The oxidation reaction at each electrode was fast in reaching the dynamic equilibrium, producing a steady-state current within almost 10 s. To calculate the sensitivity of the electrodes, calibration curves were plotted (Figure 12B), which recorded the increase in the current with each subsequent addition of DA. The linear regression equation of DA oxidation for each of N-Iron-Sp/CPE and B-Iron-Sp/CPE was obtained between 5 μM to 1.3 mM with an equation of I (µA) = 28.104 CDA (mM) + 0.7336 (R^2^ = 0.998) and I (µA) = 17.527 CDA (mM) + 0.4549 (R^2^ = 0.9981), respectively. The sensitivity of N-Iron-Sp/CPE and B-Iron-Sp/CPE was found to be 0.22 μA mM^−1^ cm^−2^ and 0.14 μA mM^−1^ cm^−2^, respectively. The remarkable electrochemical behavior of each electrode toward DA sensing is ascribed to the excellent electrocatalytic performance of crystalline Fe-oxide tightly bound to the 3D spongin scaffold. The high electrocatalytic activity, low response time of 2 s, and high sensitivity of “Iron-Spongin” are attributed to its high concentration of active sites and facile charge transfer characteristics.

The specificity of the B-Iron-Sp/CPE sensor was evaluated in the presence of possible coexisting species (sucrose, glucose, sodium chloride (NaCl), and UA). The obtained results showed that the fabricated sensor diminished the influence of possible interfering species and exhibited excellent selectivity toward DA detection. The detection of DA in human urine has received interest in medical diagnostics due to the impacts of abnormal concentrations of DA in regulating blood pressure, lipolysis, Huntington’s disease, and Parkinson’s disease. The detection of DA in human urine samples was performed using B-Iron-Sp/CPE to assess the practical applicability of the constructed DA sensor. A recovery of 93–115% was obtained for the studied sample, indicating the accuracy and reliability of the constructed sensor, which guaranteed its on-site applications.

## 3. Discussion

Lepidocrocite, as a biomineral, has been known since its discovery in the teeth of Chiton mollusc (Lowenstamm, 1967) [69]. Also, a microbial scenario of its formation, including the so-called forced biomineralization [70], is well documented in the literature. Bacterial biomineralization of lepidocrocite has been reported for diverse nitrate-reducing Fe(II)-oxidizing bacteria [71], as well as in denitrifying As(III)-oxidizing bacterium [72] under anaerobic conditions. Also, it was observed that iron oxyhydroxide crystallization could be directed during the cultivation of *Leptothrix* sp. bacterium [73]. The formation of strongly magnetic nanoscale particles due to lepidocrocite bioreduction by the iron-reducing bacterium *Shewanella putrefaciens* ATCC 8071 is described in [74]. In lithotrophic iron-oxidizing bacteria, such as *Gallionella ferruginea* or *Mariprofundus ferrooxydans*, up to 100 nm large lepidocrocite crystals nucleate on the surface of organic extracellular twisted ribbon-like stalks [75]. Maybe this phenomenon is based on the templating activity of bacterial exopolysaccharides, which are known as stabilizers of lepidocrocite. For example, iron oxyhydroxide–polysaccharide hybrid colloids with unusual pH stability of up to pH 13 are reported [76].

To the best of our knowledge, there are only two publications concerning the in vitro development of lepidocrocite-based composites using biopolymers as templates. For example, highly crystalline layers of lepidocrocite up to 125 nm large are obtained due to biomimetic mineralization of protein microtubules (MTs) with a diameter of 25 nm. It is suggested that MTs “can be used as scaffolds for the in situ production of high-aspect-ratio inorganic nanowires” [77]. In another paper, fibrillary collagen was used as a generic mineralization template for lepidocrocite [78]. The mineral phase was obtained both on and within the collagen fibrils after mixing them with Fe(OH)_2_ and the addition of poly (aspartic acid) to promote the crystallization of lepidocrocite.

Based on the previous literature data [45,46,47,79] on the interactions between marine demosponges and iron, it was possible to design a nature-inspired biomimetic method for the mineralization of iron on spongin fibers. As early as 1968 [45], the existence of crystalline iron mineralization in the spongin fibers of *Ircinia fasciculate*, *Spongia graminea*, and *S. officinalis* marine sponges was first discovered. Then, it was proven that the reddish-brown microgranules are formed of very fine crystallites of poorly organized lepidocrocite (Figure 13). It was also found that selected marine sponges grow only in the presence of iron ions [80], which are supplied to waters mainly from atmospheric sediments [81], hydrothermal vents [82], marginal sediments [83], artificial fertilization [84], groundwater discharges [85], and biocorrosion of artificial metal structures and shipwrecks [5,33,86]. The source of iron ions due to the biocorrosion of corresponding metallic constructs in seawater is crucial, especially when sponges use them as the substrate for attachment and growth [87]. Nevertheless, the mechanism of iron biomineralization on spongin fibers in nature, as well as under the laboratory conditions used in this study, is still not fully understood.

The possible mechanism of lepidocrocite formation on spongin fibers may be associated with spongin amino acid sequences, including cysteine, histidine, lysine, or tyrosine [7,8]. Functional groups derived from amino acids (e.g., –SH, –OH, –NH_2_, and –COOH) [22] and the presence of electron donor atoms (O, N, S) result in the ability to form complexes with iron ions [88]. A large group of Fe-S clusters of proteins is known; in most cases, the terminal ligands attached to iron are derived from thiol groups from cysteinyl residues [89,90,91,92]. Therefore, it is possible that cysteine/cysteine sulfur is involved in the formation of an iron-based crystalline mineral phase in spongins. Iron is a transition metal with well-known redox and ligand-binding properties [93]. It is capable of accepting and donating electrons, transitioning between the ferric (Fe^3+^) and ferrous (Fe^2+^) forms [94]. In seawater at pH 8.1, the Fe^2+^ form is rapidly oxidized to the Fe^3+^ form, so it exists mainly in the form of iron(III) oxyhydroxide, which has a very low solubility and a thermodynamically stable oxidation state [95,96,97,98]. Cornell and Schneider [99] demonstrated that in the presence of cysteine at pH 8.0, a fast transformation of non-crystalline iron(III) hydroxide into mainly crystalline lepidocrocite with a small amount of goethite occurs. Alkaline seawater conditions affect the surface chemistry of spongin—cysteine-derived thiol groups (SH–), which are converted to thiolate anions (RS–) [100]. Then, the interaction between cysteine and non-crystalline iron(III) hydroxide involves the oxidative dehydration of cysteine, which can form disulfide bonds (S–S) to produce cysteine [101,102]. There is also a concomitant reduction in some interfacial ferric sites, transforming the solid iron phase into a compound with mixed-valence Fe^2+^/Fe^3+^. This compound dissolves more readily than the starting material, and the dissolution/precipitation mechanism then leads to more thermodynamically stable iron mineral phases, such as lepidocrocite (Figure 14) [103,104,105].

Learning about the mechanism of iron mineralization in spongin fibers is essential to understanding the nature of the exceptional composite “Iron-Spongin”. It is easy and simple to prepare, and it consists of a biodegradable and renewable source—spongin. By combining both components, lepidocrocite (magneticity, stability over a wide range of temperatures (10–60 °C), pH (4.0–8.0), and spongin (3D porous structure, high thermal stability, resistance to a wide range of acids and enzymes), a nature-inspired biomaterial with many remarkable features, are created.

In this study, it was shown that such a 3D composite can be used as a sensor for neurotransmitter detection. Many methods for quantifying neurotransmitters, such as DA, are available, but most of them have their limitations [106,107,108,109]. Recently, there has been increased attention on the use of electrochemical methods for neurotransmitter analysis due to their advantages, such as high sensitivity, simplicity of analysis, fast time response, and low cost of material consumption [110]. The electroanalysis method relies on an enzymatic or enzyme-free method for detecting neurotransmitters such as DA. The main disadvantage of enzymatic biosensors is the insufficient stability of the enzymes used to develop these sensors. Their shortcomings create a real need for the development of non-enzymatic sensors. Non-enzymatic sensors generally detect chemical or biological substances through their redox activity. Electrochemical sensors based on metal oxides, such as iron, are ideal for the electroanalysis of neurotransmitters because of their simplicity, low cost, fast response, and good portability [111,112]. In their application, the electrochemical detection of specific analytes is enabled by the behavior of semiconductors, while the separation of analytes is achieved by magnetic properties. Lepidocrocite, which has magnetic properties combined with spongin, provides a large surface area and a well-developed 3D structure that seems to possess the potential for use as a DA sensor. Various magnetic iron oxide nanoparticles [113,114,115] have been proven to be excellent non-enzymatic materials for DA sensing. Previous electrochemical studies of lepidocrocite have shown its high sensitivity and selectivity in detecting DA [116]. In our study, the “Iron-Spongin” composite as a non-enzymatic electrode showed high sensitivity toward DA detection, which was attributed to the excellent electrocatalytic performance of Fe-oxide adsorbed on the unique 3D spongin scaffold. The development of “Iron-Spongin” 3D constructs in this study will stimulate experiments on their application for sodium-ion batteries, or for photocatalytic hydrogen production, where heterostructured lepidocrocite titanate-carbon nanostructures have already been used recently [117,118]. Also, such composites as potential magnetic scaffolds [119] should be investigated in the future.

## 4. Materials and Methods

### 4.1. Materials

Purified, acellular, and mineral-free spongin scaffolds of *Hippospongia communis* (Lamarck, 1814) marine demosponges were purchased from INTIB GmbH (Freiberg, Germany). InstantOcean^®^SeaSalt acquired from Spectrum Brands (Blacksburg, VA, USA) was used to prepare artificial seawater. Sodium hydroxide (analytical grade) purchased from EuroChem BGD (Tarnów, Poland) was used to prepare a 1 M (mol/L) NaOH solution. Iron powder 99.99% (with a particle size in the range of 25–100 µm) was acquired from Chempur (Piekary Śląskie, Poland). To prepare the artificial seawater, 18 g of sea salt was placed in a glass bottle and dissolved in 500 mL of distilled water. The pH of the solution was brought to pH 8.1 (the value present in natural seawater [120]) with 1 M NaOH solution. Dopamine (DA), paraffin oil, and sodium phosphate (Na_2_HPO_4_ and NaH_2_PO_4_) were purchased from Sigma-Aldrich (Burlington, MA, USA). Phosphate-buffered solution (PBS, 0.1 M, pH 6.5) was prepared using a mixture of stock solutions (NaH_2_PO_4_ and Na_2_HPO_4_) and employed as an electrolyte solution for amperometric measurements. Graphite powder was obtained from Merck (Darmstadt, Germany).

### 4.2. Samples Preparation

#### Preparation of the “Iron-Spongin” Material

A fragment of spongin scaffold weighing 1.1 g and measuring 3 cm × 6 cm × 3 cm was placed in a 500 mL bottle of artificial seawater, and 3.5 g of iron powder was added. The whole content was shaken vigorously for one minute until the entire spongin scaffold was covered with iron powder. Then, it was stored in the lab for 30 days at room temperature. Similarly, a control sample without iron powder and a control sample of iron powder alone in seawater without the presence of the spongin scaffold were prepared. After this, the obtained “Iron-Spongin” material with rusty color was placed in an ultrasonic bath (Bandelin, Berlin, Germany) for 2 h at room temperature to remove excess iron powder that did not bond to the spongin scaffold (Figure 15). The dry mass of the “Iron-Spongin” samples was measured to be 1.967 ± 0.035 g prior to and 0.708 ± 0.040 g after ultrasonic treatment.

The “Iron-Spongin” material, spongin control sample, iron from seawater alone, and iron from seawater and the presence of the sponge scaffold were then air-dried for further analysis.

### 4.3. Characterisation Techniques

#### 4.3.1. Digital Microscopy

A control sample of spongin in seawater and iron-spongin after ultrasonic treatment was observed and analyzed using an advanced imaging system consisting of a VHX-6000 digital optical microscope (Keyence, Osaka, Japan) and VH-Z20R zoom lenses (magnification up to 200×), as well as a VHX-7000 digital optical microscope (Keyence, Osaka, Japan) and VHX-E20 (magnification up to 100×) and VHX-E100 (magnification up to 500×) zoom lenses.

#### 4.3.2. Scanning Electron Microscopy (SEM) with Energy Dispersive X-ray Analysis (EDX)

For block-face analysis, the regions of interest (ROI) of TEM samples in resin blocks were trimmed using the Leica EM Trim 2 (Leica Microsystems, Wetzlar, Germany). In order to obtain a flat surface, the samples were cut with a Leica UC7 ultramicrotome using a diamond knife (Diatome, Nidau, Switzerland).

The samples were mounted on a heavy metal-free Al-SEM-carrier (co. PLANO, Wetzlar, Germany) with adhesive conductive carbon tape (Spectro Tabs, TED PELLA INC, Redding, CA, USA) and coated with carbon (5.0 nm thickness) under vacuum (CCU 010 HV-Coating Unit, Co. Safematic GmbH, Zizers, Switzerland).

The samples were analyzed using a field emission scanning electron microscope (SEM, MERLIN^®^ VP Compact, Co. Zeiss, Oberkochen, Germany) equipped with an energy-dispersive X-ray (EDX) detector (XFlash 6/30, Co. Bruker, Berlin, Germany). Representative areas or defined lines of the samples were analyzed and mapped for elemental distribution based on the EDX-spectra data using QUANTAX ESPRIT Microanalysis software (version 2.0, Berlin, Germany) SEM images were taken from selected regions (the conditions are shown in the figures).

Comparative SEM-EDX analyses of the control sample and iron-spongin after ultrasound treatment were carried out using a scanning electron microscope (Quanta 250 FEG; FEI Ltd., Brno, Czech Republic) correlated with an energy-dispersive X-ray spectrometer (EDX Team Software) to determine the elemental composition and surface morphology of the samples studied.

Moreover, SEM and supplementary EDX measurements were carried out using a low-vacuum scanning electron microscope, JEOL JSM-6610LV, with a LaB6 cathode, which was also equipped with an energy-dispersive X-ray spectrometer (10 mm^2^ Silicon Drift Detector (SDD) X-Flash 6|10, Bruker Co., Berlin, Germany).

#### 4.3.3. High-Resolution Transmission Electron Microscopy (HR-TEM)

Conventional TEM analysis was carried out using the FEI Tecnai F30-G^2^ with Super-Twin lens (FEI) with a field emission gun at an acceleration voltage of 300 kV. The point resolution amounted to 2.0 Å, and the information limit was about 1.2 Å. The microscope was equipped with a wide-angle slow-scan CCD camera (MultiScan, 2k × 2k pixels; Gatan Inc., Pleasanton, CA, USA).

#### 4.3.4. Transmission Electron Microscopy (TEM)

Selected fragments of “Iron-Spongin after ultrasonic treatment” were placed in distilled H_2_O for one night at room temperature (RT). Then, they were dehydrated in an ethanol series from 30% to 100% at RT and embedded in Araldite (Sigma-Aldrich, Burlington, MA, USA) epoxy embedding media according to the manufacturer’s instructions. Ultra-thin sections (60–70 nm) were cut with an Ultramicrotome PowerTome XL (Boeckeler Instruments, Inc., Tucson, AZ, USA) equipped with a Druker International b.V (Amsterdam, the Netherlands) 45 diamond knife, double-stained with UranyLess (EMS), lead citrate, and lead citrate. Ultrathin sections were studied under Tecnai G2 20 TWIN (FEI Company, Alhambra, CA, USA) transmission electron microscope with an acceleration voltage of 200 kV.

#### 4.3.5. Fourier-Transform Infrared Spectroscopy

FTIR spectra of the control and obtained samples were recorded using a Nicolet iS50 spectrometer (Thermo Fisher Scientific Co., Hillsboro, OR, USA). Each measurement was performed using a built-in attenuated total reflectance (ATR) accessory. The analysis was carried out in the wavelength range of 4000–400 cm^−1^.

#### 4.3.6. X-ray Diffraction

The X-ray studies of the examined materials were performed using a powder diffractometer (SmartLab Rigaku, Tokyo, Japan) with a Cu K-alpha X-ray tube, in the range of 3–80 (2 theta), scan step 0.01, and scan speed 4°/min.

#### 4.3.7. Thermogravimetric Analysis

Thermogravimetric analysis (TG/DTG) was performed on a TGA/DSC1 Star Systemanalyzer (Mettler Toledo, Columbus, OH, USA) Measurements were carried out at a heating rate of 10 °C/min under nitrogen flow conditions (60 mL/min) in the temperature range of 30–700 °C.

#### 4.3.8. Confocal Micro X-ray Fluorescence (CMXRF)

CMXRF measurements were performed with a modified commercial MXRF spectrometer (M4 TORNADO) by Bruker Nano GmbH, Berlin, Germany, which was equipped with a 30 W Rh-microfocus X-ray tube (50 kV, 600 µA), a polycapillary full lens in the excitation channel for X-ray focusing, and a 30 mm^2^ silicon drift detector (SDD). Due to the modification, a polycapillary half lens was installed in the detection channel before a 60 mm^2^ SDD. The confocal arrangement of both lenses resulted in a defined probing volume, providing three-dimensional resolved element analysis by lateral movement of the sample with an xyz-motorized sample stage. The calibration of the optics alignment was realized by the precise movement of the second lens by piezo actuators and tracking the signal intensity of a 2 µm thick Cu foil.

The CMXRF measurements were performed within a total sample volume of 500 × 500 × 500 µm^3^ and a global step size of 5 µm. A spot measurement time of 10 ms was utilized with five measurement cycles, resulting in a measurement time of 50 ms for each point and an overall measurement time of approximately 63 h. Additionally, with respect to the presence of light elements in the spongin samples, a vacuum of 20 mbar was applied for all the measurements.

For the first data evaluation of the 101 generated xy area mappings at varying z positions, the spectrometer corresponding software was utilized, providing the impulse count values for the element signals Si-Kα (1.740 keV), S-Kα (2.307 keV), Ca-Kα (3.691 keV), I-Lβ (4.239 keV), Fe-Kα (6.397 keV), and Br-Kα (11.902 keV). Due to the physical properties of the lenses used, quite different probing volume sizes need to be considered for the different fluorescence energies of the element lines. For the utilized setup, the probing volume sizes were calculated as a function of the energy by calibrating the characteristics of the spectrometer parameters [121]. Hereby, the following probing volume z-sizes can be expected in approximation: Si-Kα (77.2 µm), S-Kα (69.0 µm), Ca-Kα (55.6 µm), I-Lβ (51.8 µm), Fe-Kα (42.0 µm) and Br-Kα (31.4 µm).

The exported measurement datasets (containing information about the location coordinates x and y and the signal count values) were then further processed using in-house software (applied in references [122,123]), providing tools like the normalization of the xy mappings to a global signal maximum, generating RGB color-coded images and signal noise correction. The final stacking of the two-dimensional distribution datasets was carried out with the Python application Mayavi, achieving three-dimensional distribution images. For the three-dimensional reconstruction of the element distributions (Si (cyan), S (yellow), Ca (blue), I (magenta), Fe (red), and Br (green)) a volume module was used in combination with light and shade calculations for better visibility of the three-dimensional structure. Due to the small size of the sponge structure (~30 µm) compared to the probing volume sizes (≥31.4 µm), weak signal values were excluded from the volume rendering by setting the RGB alpha value to zero.

Due to the properties of natural samples (varying density, elemental composition, and absorption due to the 3D structure) and different physical behaviors of the observed elements (fluorescence yield, sensitivity, and concentration), different alpha values were utilized for the volume reconstruction of each element and sample. Hereby, data points within the range of 1 to 10% of the global maximum count value were excluded, aiming for a less cluttered representation of the 3D elemental distributions. Therefore, the volume reconstructions depict only a qualitative approximation of the 3D elemental distribution. Further data processing is needed for the correction of the influences of probing volume size and absorption effects. Since these samples exhibit a quite complex three-dimensional structure and composition, the feasibility of these complex reconstruction tasks (both qualitatively and quantitatively) needs to be addressed in future work.

#### 4.3.9. Magnetic Properties

The magnetic properties of the obtained “Iron-Spongin” material were tested using a neodymium magnet with a pull force of 192 N, purchased from Mistral, Jaworzno, Poland.

### 4.4. Dopamine Detection

For the sensor preparation, modified carbon paste electrodes (CPE) were fabricated by grinding graphite, paraffin oil as a binder, and a modifier in a mortar with a ratio of 65:15:20 (*w*/*w*/*w*) and a grinding time of 40 min. The components were homogenized to form a paste, which was then pressed into a holder with an inner diameter of 4 mm.

Amperometric measurements were carried out using a PalmSens 4 electrochemical analyzer with the software PSTrace 5.8 (PalmSens BV, Houten, the Netherlands) and a three-electrode setup including modified CPE as the working electrode, Ag/AgCl (3 M KCl) electrode as the reference, and a platinum wire as the counter electrode. The amperometric response of the different modified CPEs for the successive addition of DA in 0.1 M phosphate buffer pH 6.5 was recorded at a potential of 0.25 V.

## 5. Conclusions

This study focused on a better understanding of the interaction between biomaterial spongin and iron ions in marine environments due to biocorrosion, which led to the occurrence of the biomineral lepidocrocite. For this purpose, a biomimetic approach for the creation of a new lepidocrocite-containing 3D spongin scaffold using artificial seawater and iron powder under laboratory conditions at 24 °C is described for the first time. This simple method allowed obtaining a new composite called “Iron-Spongin”. The limiting factors such as the concentration of iron ions, pH, and temperature should be studied in the future with the aim of finding optimal parameters for the development of functional lepidocrocite-based 3D composites on a large scale.

The discovery of rusty bath sponges in the industrial production of marine sponges, from both open ocean colonies and those grown in marine culture, is not uncommon. On the contrary, rusty sponges are found in mass quantities (Figure 13) and are rejected by the respective companies due to a lack of demand or use for traditional cosmetic purposes.

However, our work shows the possibility of further application of such rusty sponges in biomimetics and materials science. Consequently, the strategy for the use of these specific sponges must be changed drastically. This opens a way for the sustainable and correct use of sponges without the presence of substandard biomaterials. Intriguingly, technologies have been developed to grow sponges under marine ranching conditions on reinforced iron pins or plates to create iron-containing composites as functional materials.

## Figures and Tables

**Figure 1 marinedrugs-21-00460-f001:**
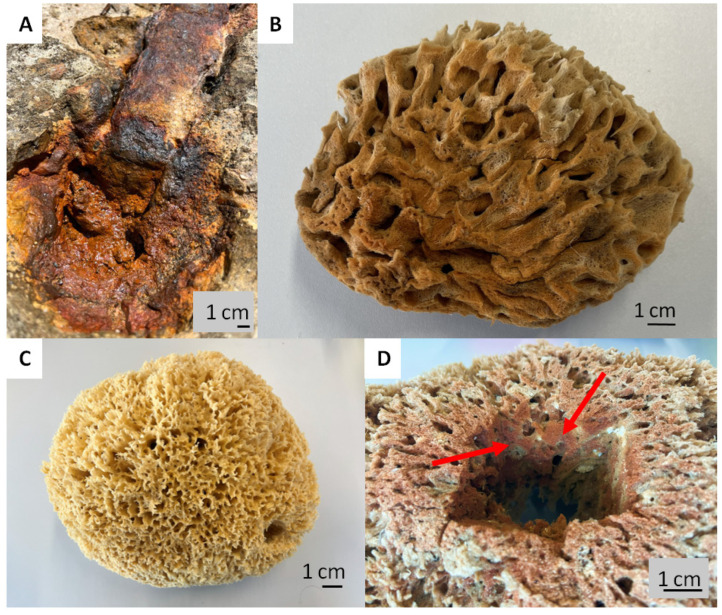
Corrosion of artificial iron-based tools after contact with seawater: (**A**) remains to be the main source of iron ions in the aquatic environment surrounding the bath sponges. This leads to the development of a well-visible rusty coloration (**B**) due to the presence of the lepidocrocite mineral phase tightly attached to the organic spongin. A natural skeleton isolated from the marine demosponge *Hippospongia communis* growing with the absence of iron ions (**C**) is yellowish in color. This kind of iron-based biomineralization is also detectable deep within the sponge skeleton ((**D**), arrows).

**Figure 2 marinedrugs-21-00460-f002:**
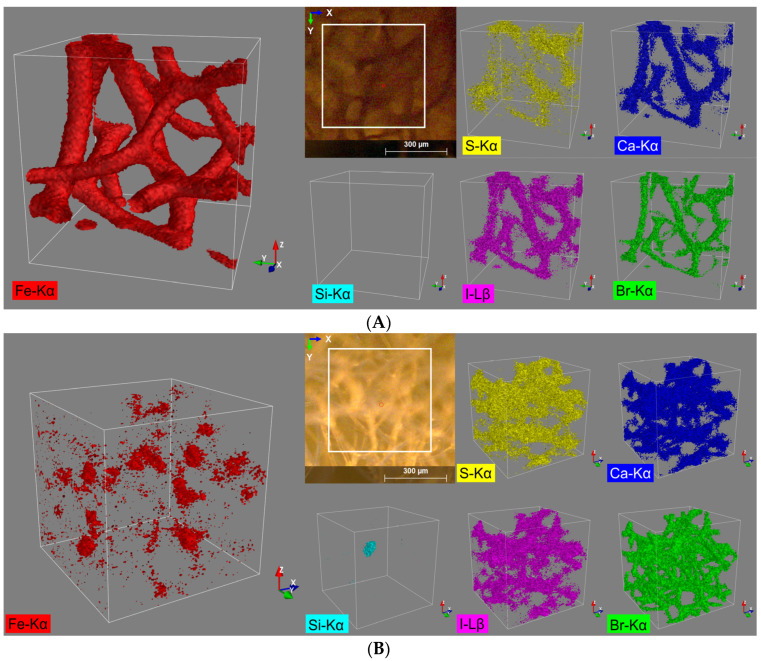
3D distribution images of the elements S (Kα), Ca (Kα), Fe (Kα), Br (Kα), and I (Lβ) within an analysis volume of 500 µm × 500 µm × 500 µm of the (**A**) spongin with naturally formed lepidocrocite and (**B**) pure spongin (control).

**Figure 3 marinedrugs-21-00460-f003:**
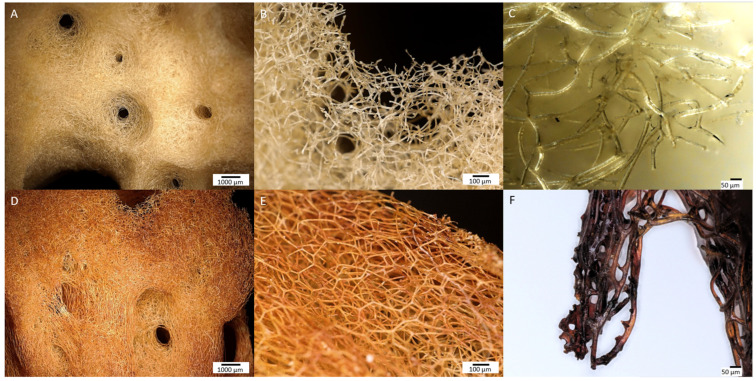
Digital microscopy imagery of two different samples of spongin from lower to higher magnifications (see the scale bars). (**A**–**C**) Control sample of spongin isolated from *H. communis* demosponge growing in a Fe-free environment remain to be yellowish in color. This biomaterial known as commercial, or bath sponge, found broad applications in human life. However, the same sponge material after induced corrosion of iron powder in artificial seawater for 30 days (**D**–**F**) becomes a rusty color that is still stable even after 2 h of ultrasonic treatment at 24 °C.

**Figure 4 marinedrugs-21-00460-f004:**
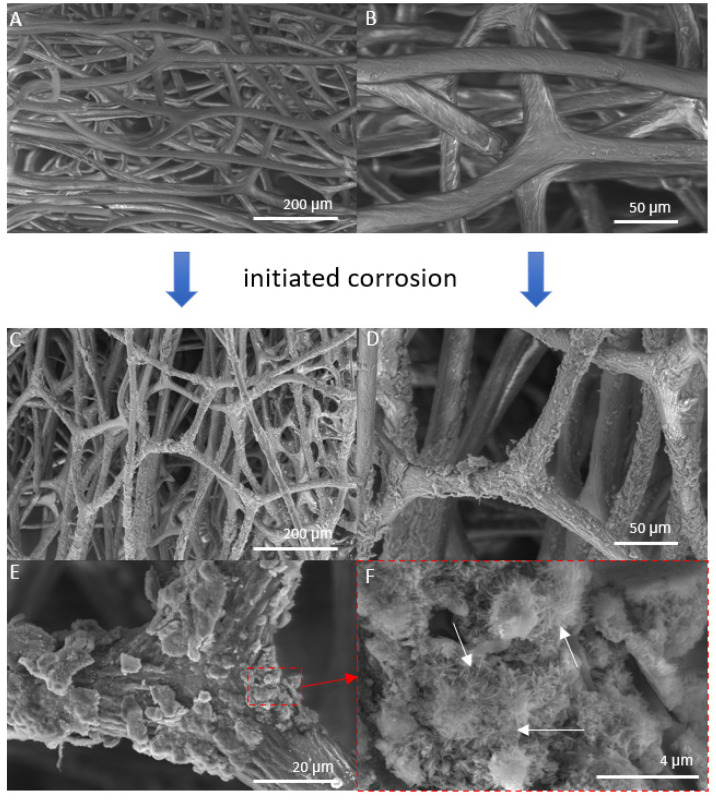
SEM images of iron-free spongin fibers with both lower (**A**) and higher (**B**) magnifications (scale bars represent 200 µm and 50 µm, respectively) (see also Figure 2A–C) drastically differ from that obtained after “Iron-Spongin” 3D composite, where the formation of crystalline phase (**C**–**F**) (scale bars represent 200 µm, 50 µm, 20 µm, and 4 µm, respectively) remains to be well visible even after ultrasonic treatment. (**F**) Arrows show needle-like crystals.

**Figure 5 marinedrugs-21-00460-f005:**
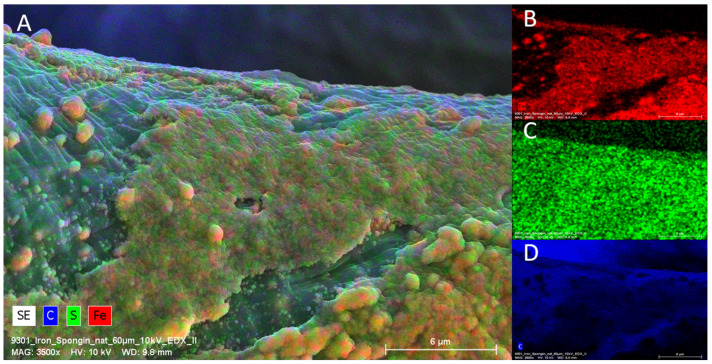
EDX Analysis (elemental mapping) of a single fiber of an “Iron-Spongin after ultrasound treatment” sample. Clearly visible is the presence of Fe and S on the surface of the scaffold strain (**A**). Fe is predominantly deposited in the crust-like structure (**B**), whereas sulfur is more or less equally distributed over the surface (**C**). C is present on the whole sample due to the organic compounds of the sample and of carbon coating used for SEM (**D**).

**Figure 6 marinedrugs-21-00460-f006:**
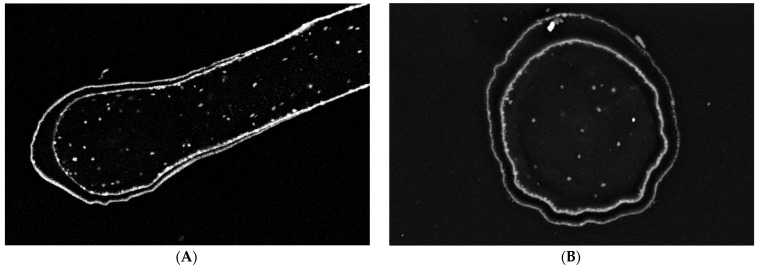
Block face images of a single spongin fiber: longitudinal section (**A**) and cross-section (**B**). In both images, two different layers are distinguishable. According to the high contrast given in these layers, the presence of elements with higher atomic numbers—combined with elements origin from biological tissue—is most likely. Bars represent 10 µm.

**Figure 7 marinedrugs-21-00460-f007:**
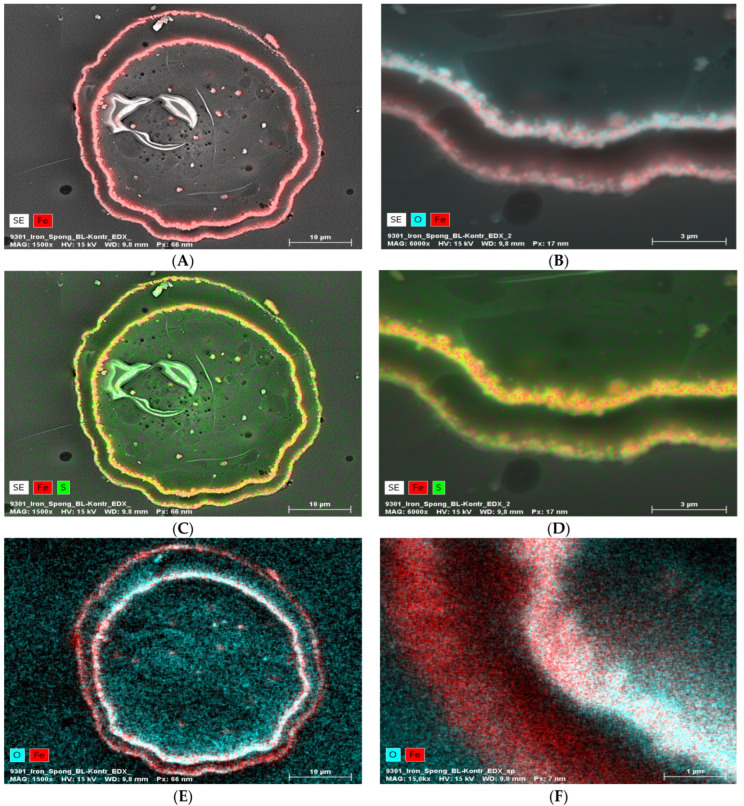
EDX analysis (elemental mapping) of a block face image cross-section of a single “Iron-Spongin after ultrasound treatment” fiber, whereby the element contents are colored in red for iron, blue for oxygen, and green for sulfur (in the case of red and green overlapping, a bright yellow color is observed). The presence of Fe and O in the two layers (**A**,**B**,**E**,**F**) is clearly visible. Sulfur seems more or less equally distributed over the cross-section (**C**,**D**). The inner layer seems to be higher in Fe and O combined with the outer layer (**B**,**E**,**F**).

**Figure 8 marinedrugs-21-00460-f008:**
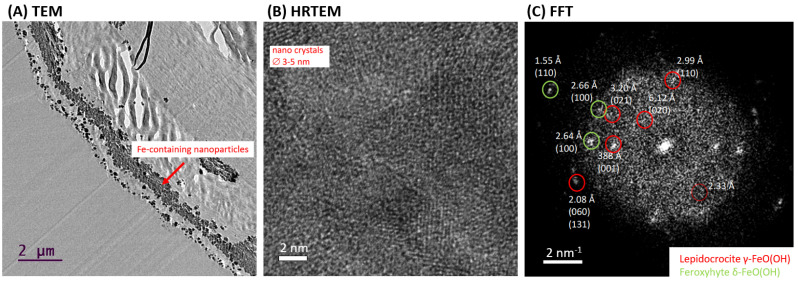
TEM overview (**A**) and high-resolution TEM (**B**) of Fe-containing nanoparticles on a selected nanofiber of “Iron-Spongin” composite investigated after ultrasound treatment. Calculated fast Fourier transform (FFT) with measurement of interplane separations indicating the occurrence of lepidocrocite and possible minor phase of feroxyhyte (**C**).

**Figure 9 marinedrugs-21-00460-f009:**
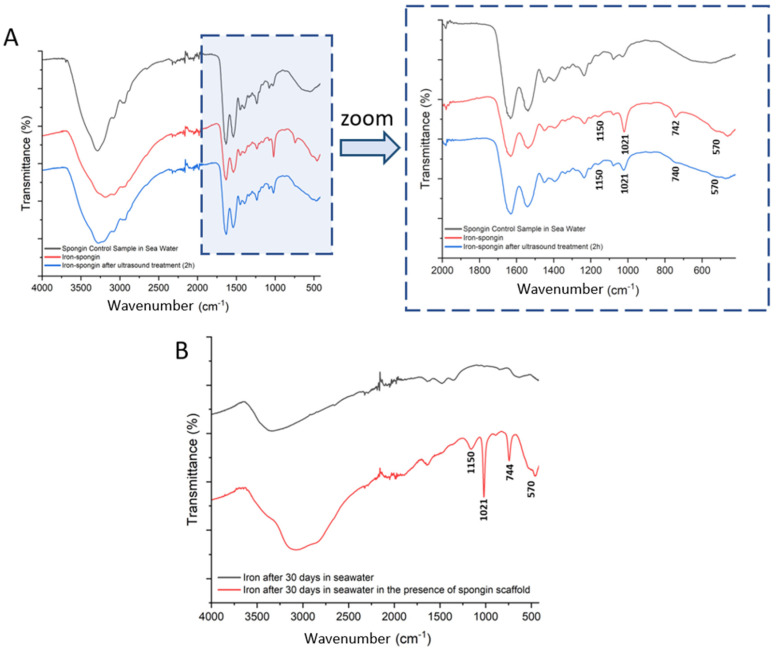
FTIR spectra of the samples: (**A**) spongin control sample and “Iron-Spongin after ultrasound treatment” along with an approximation in the range of 2000–420 cm^−1^; (**B**) iron powder after 30 days insertion into seawater with and without the presence of spongin scaffold.

**Figure 10 marinedrugs-21-00460-f010:**
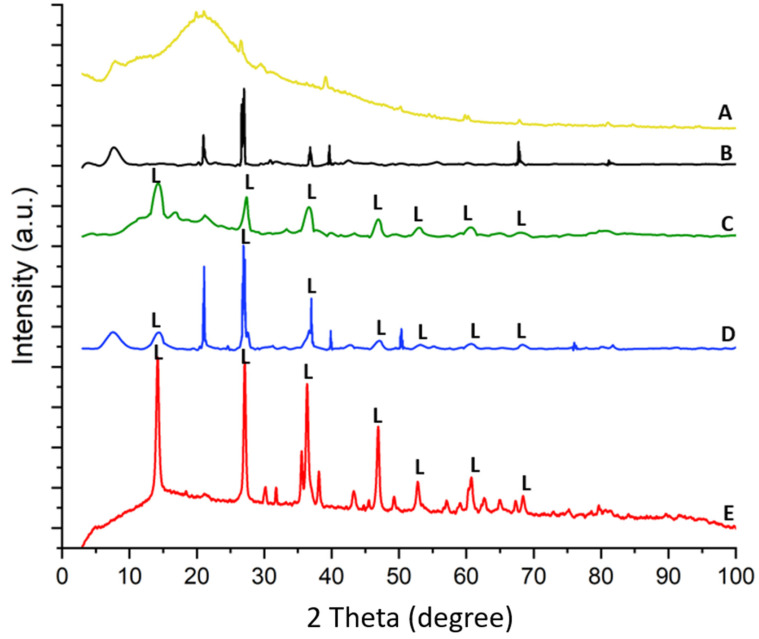
XRD patterns of (**A**) spongin, (**B**) spongin control sample after placement into seawater, (**C**) “Iron-Spongin”, (**D**) “Iron-Spongin after ultrasound treatment”, and (**E**) iron powder after placement in the seawater in the presence of spongin.

**Figure 11 marinedrugs-21-00460-f011:**
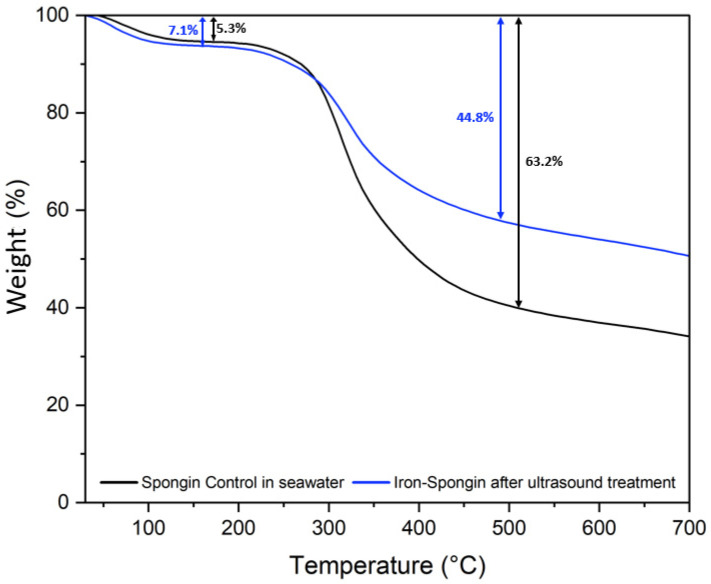
Thermogravimetric (TG) curves of spongin control sample in seawater and the “Iron-Spongin after ultrasound treatment”.

**Figure 12 marinedrugs-21-00460-f012:**
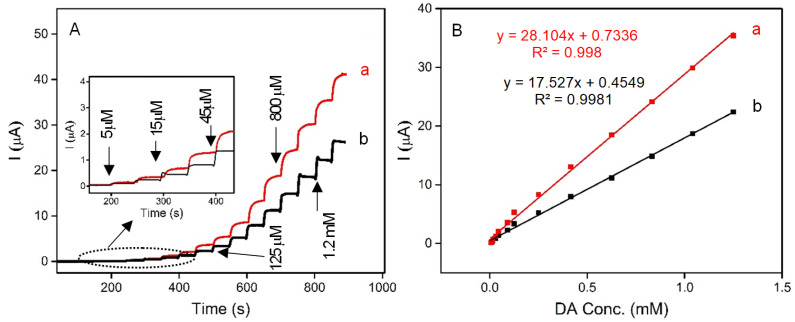
(**A**) Amperograms recorded in 0.1 M phosphate buffer pH 6.5 with the successive addition of DA (5 μM to 1.3 mM) at (a) N-Iron-Sp/CPE, and (b) B-Iron-Sp/CPE. (**B**) Calibration curve for the linear response of current vs. DA concentration.

**Figure 13 marinedrugs-21-00460-f013:**
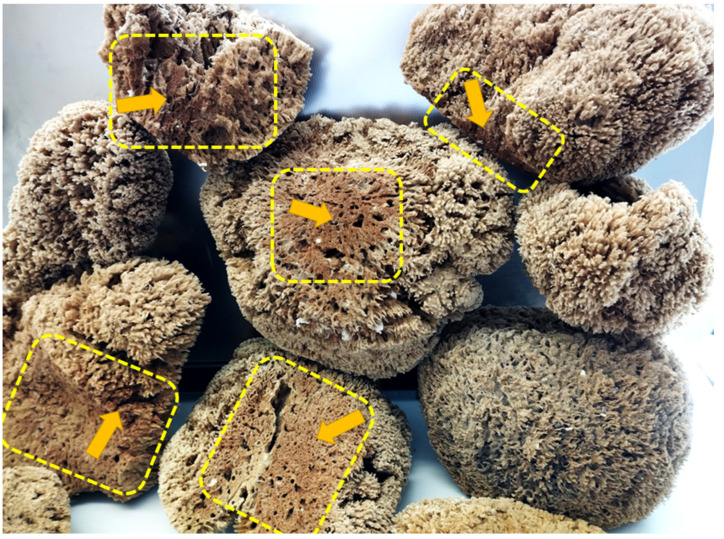
Rusty bath sponges represent a potential source of naturally occurring iron oxide-based 3D constructs, which can be useful in bioinspired material science and biomimetics. Well-defined lepidocrocite-containing locations within the sponge skeletons are marked in yellow.

**Figure 14 marinedrugs-21-00460-f014:**
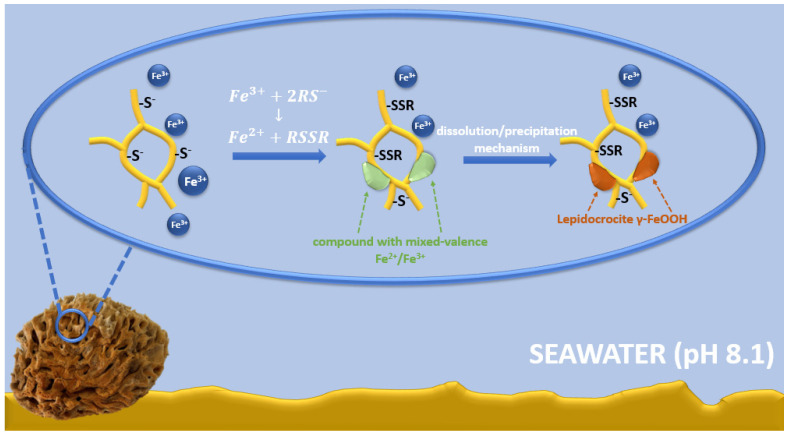
Schematic representation of the possible mechanism of lepidocrocite formation on spongin fibers.

**Figure 15 marinedrugs-21-00460-f015:**
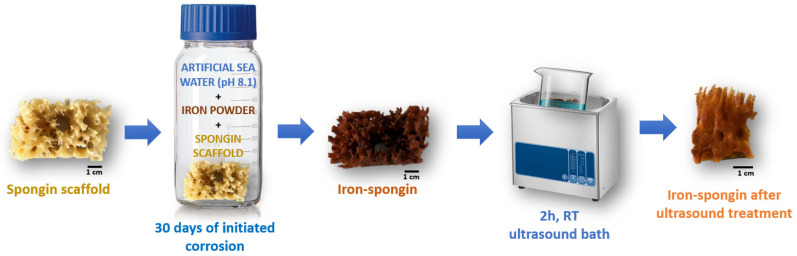
Schematic overview of the preparation of “Iron-Spongin” material using iron powder in artificial seawater with pH 8.1 at 24 °C.

**Table 1 marinedrugs-21-00460-t001:** CMXRF measurements (maximum voxel counts) for the elements identified in the samples.

Sample/Signal Count Rates	Fe-Kα	Br-Kα	Ca-Kα	S-Kα	I-Lβ	Si-Kα
Spongin Fe-natural	243.0	39.0	31.0	27.0	15.0	-
Spongin pure (control)	18.0	19.0	19.0	18.0	19.0	12.0

## Data Availability

Not applicable.

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
