# Peer review of "Spongin as a Unique 3D Template for the Development of Functional Iron-Based Composites Using Biomimetic Approach In Vitro"

_marinedrugs, 2023, doi:10.3390/md21090460_

Round 1
Reviewer 1 Report
In this study, a biomimetic method for development of lepidocrocite on a spongin scaffolds using artificial seawater was illustrated. It was presented that the reaction in an artificial seawater environment between spongin template and iron ions leads to the formation of a new 3D composite material called “Iron-Spongin” that resemble the size and shape of original sponge skeleton, and discussed the corresponding mechanism of possible formation of crystalline lepidocrocite on spongin. I think there are some questions should be noticed:
1. The abstract must contain an introduction, objectives, methodology, findings, and conclusions. Please rewrite the abstract.
2. The keywords should complement and not repeat the title (remember that the aim of keywords is to increase visibility of your article in the databases, therefore more general words might be relevant, or words that describe the work but you did not manage to include in the title); please change some of the words.
3. For Figure 1, please list the pictures in order. For Figure 3, please explain the difference or magnifications of Figure 3A-C, and Figure 3D-F, as well as Figure 4 A, B and Figure 4 C-F. In Figure 3C and 3F, there should have a space between 50 and µm. in Figure 4F, what does the arrow point to? It should be indicated in the Figure caption.
4. In Figure 10, the title of the X-axis should be “Wave number”.
5. In Figure 11, the legend is a repeat of the figure caption, I think the legend can be deleted.
6. Line 319, “In Figure 13 can be observed”, it looks like the grammar is not right, please revise this sentence.
7. Too many figures and Tables, I think you can combine some figures, or delete some that not very import or put them into supplements.
8. In Materials and Methods section, please add the name of city or town of the instrument, such as line 518, line 564, line 569, please check throughout this section and add them all.
9. After Discussion, please add a Conclusion section, which should highlight major findings, linked to your objectives and to give a take home message to the readers. Therefore, it should answer the following questions: What is new? Why is it important? What are the limitations and potential consequences?
Reviewer 2 Report
The paper presented by A. Kubiak et al. presents an interesting point of view regarding marine sponges. The presented results are important for the scientific community. However, despite the fact that there are numerous results presented, the work is difficult to follow. In my opinion, authors need to organize their results better, highlight their contribution/novelty of the work and be more concise. At the same time, there are too many references (almost 140). Perhaps the authors can select the references and keep only the relevant works.
Minor notes: In the figure captions please avoid writing sentences, present the figures in the order A, B and C. The description should be in the text.
Please also check the English language, sometimes the sentences are not clear.
Reviewer 3 Report
Manuscript ID: marinedrugs-2527227. Title: Spongin as unique 3D template for development of functional Iron-based composites using biomimetic approach in vitro
Recommendation: minor revision.
This is an interesting work in which the authors report a biomimetic approach for the development of a lepidocrocite-containing 3D spongin scaffold under laboratory conditions using artificial seawater and iron powder. The designed composite is characterized through various characterization techniques and the corresponding mechanism of its formation as well as its application as a sensor for the detection of dopamine is presented and discussed.
Overall, the manuscript is well-written, meets the criteria of the Journal and the Special Issue and could be published after minor revision.
Some remarks and corrections
1. Introduction, Line 52. due to ability – due to its ability
2.1. Confocal Micro X-ray Fluorescence (CMXRF). Why did the authors not study the Iron-Spongin scaffold before and after ultrasound treatment as well?
2.3. Scanning Electron Microscopy (SEM) with Energy Dispersive X-Ray Analysis (EDX). In EDX analysis is it the scaffold after the ultrasound treatment that was examined? It is not clear in the figure captions (figures 5-8) if it is the scaffold before or after ultrasound treatment.
2.5. Fourier-Transform Infrared Spectroscopy. Line 279. The effect of spongin scaffold on iron corrosion in seawater was also investigated. Perhaps the authors should comment more on the effect of spongin scaffold on iron corrosion in seawater.
2.6. X-ray diffraction. Figure 11. The x-axis title maybe should be better to be presented as 2 Theta (degree)
2.7. Thermogravimetric analysis. Figure 12. Y-axis title. Is it Mass-loss (%) or should be Mass (%) or Weight (%)
2.8. Magnetic properties. Figure 13. Is it the Iron-spongin scaffold or the Iron-spongin after ultrasound treatment?
2.9. Dopamine Detection. Which Iron-Spongin scaffold did the authors study? Was it the one after ultrasound treatment?
In many cases, it is not clear to the reader which Iron-Spongin scaffold the authors have studied in the different characterization techniques (the one before or after ultrasound). Perhaps it would be better if the authors named somehow different the scaffold after ultrasound treatment (e.g. Iron-Spongin ULS). Also, why in some techniques both scaffolds before and after ultrasound treatment have been studied (e.g. FTIR, XRD) while in others only the scaffold after the ultrasound treatment was examined (e.g. SEM, EDX, TEM, TGA)?
3. Discussion. Line 385. Also microbial scenario of it formation - Also a microbial scenario of its formation
Line 421. The possible mechanism of lepidocrocite.. Is it the same mechanism described in a previous work of the authors? (Citation 7) Why should be reported again here?
4.2.1. Preparation of the “Iron-Spongin” material. What was the volume of water? The concentration of spongin and iron powder was based on what criteria? Did the authors examine different concentrations and if yes what were the results?
The mass of the “Iron-Spongin” samples has been measured as.. - Was it a dry-mass?
The Section 5 title should be Conclusions not Outlook.
Minor text editing
Round 2
Reviewer 1 Report
For the Conclusion section, still needs better to focus on What is new? Why is it important? and limitations? Normally, there is no need to cite references, tables and figures in the Conclusion section. One paragraph is fine for Conclusion. Please organize this section better.
Author Response
Thank you again for this critical remark. We made corresponding changes. However, we decide still to represent Fig.15 in this chapter as very informative with respect to future research and better understanding of the matter.